# Effect of Ca Deoxidation on Toughening of Heat-Affected Zone in High-Strength Low-Alloy Steels after Large-Heat-Input Welding

**Yinhui Zhang** [1] , **Jian Yang** [1,*], **Hailong Du** [1], **Yu Zhang** [2] **and Han Ma** [2]

[1] State Key Laboratory of Advanced Special Steel, School of Materials Science and Engineering, Shanghai University, Shanghai 200444, China

[2] Institute of Research of Iron and Steel, Sha-Steel, Zhangjiagang 215625, China

\* Correspondence: yang_jian@t.shu.edu.cn; Tel.: +86-021-6613-6580

**Abstract:** Large-heat-input welding can effectively increase the efficiency and reduce the cost of manufacturing a super-large container ship for marine trade worldwide with thick, high-strength low-alloy (HSLA) steel plates; however, it significantly degrades the toughness of the welding heat-affected zone (HAZ). This paper describes the effect of Ca deoxidation on the impact toughness of simulated coarse-grained HAZs (CGHAZs) in HSLA steels after large-heat-input welding at 400 kJ cm$^{-1}$. The average impact energy of the CGHAZ increases with an increase in Ca content; in particular, the energy of the steel with 25 ppm Ca content is satisfactorily high, owing to the uniform and fine prior austenite grains. In contrast, the grains in the CGHAZs of the steels with relatively low Ca contents are not uniform, leading to large test variabilities at −20 °C. Failure analyses reveal that the major and secondary cracks preferentially propagate along the coarse and brittle grain boundary ferrite (GBF), but their propagation is retarded by the fine and interlocking intragranular acicular ferrite (IAF) as the grain size decreases. It is therefore recommended to increase the Ca content to about 25 ppm during the deoxidation of steelmaking to improve HAZ toughness by forming fine and uniform prior austenite grains and IAF within grains.

**Keywords:** impact toughness; large-heat-input welding; heat-affected zone; high-strength low-alloy steel; Ca deoxidation

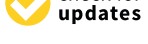



## 1. Introduction

Thick, high-strength low-alloy (HSLA) steel plates are very attractive for manufacturing large structures due to their excellent combination of strength, toughness and weldability [1]. The introduction of large-heat-input welding technology can significantly improve production efficiency, but it also deteriorates the toughness of the heat-affected zones (HAZs) of steels, particularly coarse-grained HAZs (CGHAZs) adjacent to weld beads [2,3]. A low HAZ toughness indicates a high risk of cold cracking, and it restricts the use of thick steel plates for the production of super-large container ships [4].

Numerous efforts are currently focused on the refinement of the HAZ microstructure using oxide metallurgy technology, thereby increasing HAZ toughness after large-heat-input welding [5–12]. Xu et al. reported that the addition of Mg to shipbuilding steel can dramatically prevent the growth of prior austenite grains and accelerate the formation of fine and interlocking intragranular acicular ferrite (IAF) and polygonal ferrite (PF), ultimately increasing HAZ toughness after welding at 400 kJ cm$^{-1}$ [13,14]. Wang et al. reported that the combined deoxidation of Ti and Zr forms fine and complex inclusions and promotes the nucleation of IAF on them, leading to a much better HAZ toughness than C-Mn steel without the addition of Ti or Zr after welding at 100 kJ cm$^{-1}$ [15]. Yu et al. reported that adding a trace amount of Y can change inclusions into Y-bearing oxy-sulfides and induce IAF nucleation, which promotes HAZ toughness accordingly [16].

Besides the positive effects of strong deoxidants, such as Mg, Ti, Zr and rare-earth elements, Ca is also a strong deoxidant and is commonly used in steelmaking [17]. Choudhary

et al. developed a thermodynamic model to predict the formation of oxy-sulfide arising out of competitive reactions between [O], [S] and [Ca] in Al-killed steel [18]. Kato et al. developed an oxide metallurgy technology via Ca deoxidation and found that it can form CaO-CaS oxy-sulfides and improve the fine dispersion of TiN particles [19]. Our previous studies summarized that Ca oxide metallurgy improves the precipitation of sub-micron scale TiN particles, which can prevent grain growth, and CaO-CaS oxy-sulfides, which can induce the formation of IAF after large-heat-input welding at 400 kJ cm$^{-1}$ [20–22]. It should be mentioned that Ca oxide metallurgy generally refers to deoxidation by Ca rather than Al during steelmaking, different from the traditional Ca treatment that transforms solid $Al_2O_3$ into liquid $Al_2O_3$–CaO to prevent nozzle clogging [22]. However, very limited research on Ca oxide metallurgy has been conducted, resulting in the toughening mechanism of the HAZs of HSLA steels by Ca deoxidation remaining unclear. Moreover, the effect of ultra-large-heat-input at 400 kJ cm$^{-1}$ on the microstructure and toughness of HAZs has not been thoroughly investigated, including those of ferritic and bainitic HSLA steels [23–26]. Therefore, the objective of the current paper is to investigate the toughening mechanism of HAZs after large-heat-input welding at 400 kJ cm$^{-1}$ as a function of Ca addition. CGHAZ specimens were obtained by varying the Ca content during steelmaking, followed by welding simulations using a Gleeble 3800 thermal–mechanical physical simulator and Charpy impact tests at −20 °C. The main focuses are (I) the uniformity of the prior austenite grains to clarify the variability of the impact tests and (II) crack propagation to explore the fracture mechanism of CGHAZs. The generated data and understanding will help to improve the HAZ toughness of this type of HSLA steel using the Ca deoxidation method.

## 2. Materials and Methods

A series of HSLA steels were melted in a 50 kg vacuum induction furnace under an Ar atmosphere and deoxidated by different Ca additions. The chemical compositions of these steels were quantitatively measured, and they are listed in Table 1, denoted as 2Ca, 11Ca, 18Ca and 25Ca steels depending on the Ca content. The steel ingots were rough-rolled at >930 °C and finish-rolled at >800 °C under the thermo-mechanical control process (TMCP), with a final plate thickness of 50 mm.

**Table 1.** Chemical compositions of the HSLA steels with different Ca contents (mass%).

| Steels | C | Si | Mn | P | S | Al | N | Ti | O | Ca |
|--------|------|-----|-----|-------|-------|-------|--------|--------|--------|--------|
| 2Ca | 0.08 | 0.2 | 1.5 | 0.006 | 0.004 | 0.001 | 0.0034 | 0.0093 | 0.0018 | 0.0002 |
| 11Ca | 0.08 | 0.2 | 1.5 | 0.009 | 0.007 | 0.004 | 0.0048 | 0.013 | 0.0019 | 0.0011 |
| 18Ca | 0.08 | 0.2 | 1.5 | 0.007 | 0.004 | 0.006 | 0.0044 | 0.0096 | 0.0012 | 0.0018 |
| 25Ca | 0.08 | 0.2 | 1.5 | 0.007 | 0.005 | 0.007 | 0.0037 | 0.011 | 0.0019 | 0.0025 |

A Gleeble 3800 thermal–mechanical physical simulator (Dynamic Systems Inc., Poestenkill, NY, USA) was used to simulate electrogas arc welding with a heat input of 400 kJ cm$^{-1}$ for the 50 mm-thick steel plates. Electrogas arc welding with a heat input of 400 kJ cm$^{-1}$ is a one-pass automatic welding method, which can significantly increase welding efficiency and reduce costs [27]. Figure 1 shows schematic images of the specimens used for the large-heat-input welding simulations and Charpy impact tests. The specimens were processed into a size of 11 × 11 × 71 mm$^3$, with a thermocouple welded at the middle part to control the thermal cycle temperature. Figure 2 shows the thermal cycle for the simulation of the electrogas arc welding with a heat input of 400 kJ cm$^{-1}$. The specimens were heated to a peak temperature of 1400 °C at a rate of 479 °C s$^{-1}$ and held for 3 s. Subsequently, they were cooled to 800 °C at a rate of 3.4 °C s$^{-1}$, followed by cooling from 800 to 500 °C for 385 s ($t_{8/5}$) and continuous cooling to room temperature at 0.2 °C s$^{-1}$. The specimens were processed into a size of 10 × 10 × 55 mm$^3$, with a V-notch at the position of the thermocouple after the welding simulations, and they underwent the Charpy impact tests at −20 °C in accordance with the ASTM E23 standard.

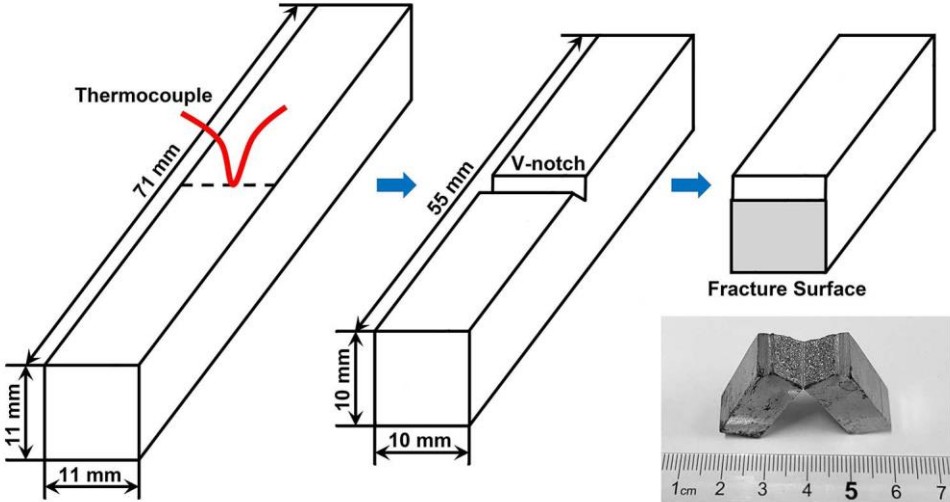

**Figure 1.** Schematic images of the specimens used for large-heat-input welding simulations and Charpy impact tests.

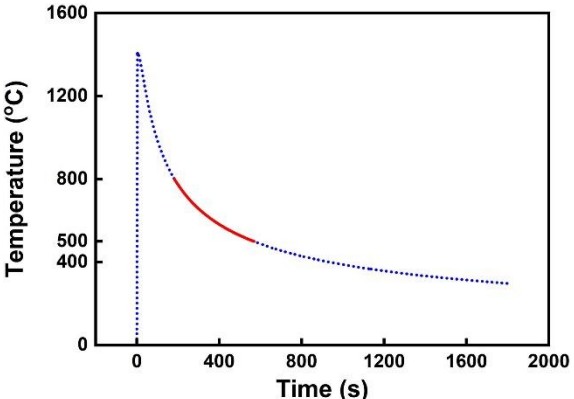

**Figure 2.** Thermal cycle for the simulation of the electrogas arc welding with a heat input of 400 kJ cm$^{-1}$ for the 50 mm-thick steel plates.

The specimens of the simulated HAZs parallel to the crack surface were mechanically polished and etched using a 4% nitric acid solution for microstructural characterization by an optical microscope (OM, DM 2700 M, Leica Microsystems, Wetzlar, Hessian, Germany) after the Charpy impact tests. The fracture surfaces of the specimens were examined using a scanning electron microscope (SEM, EVO 18, Carl Zeiss AG, Oberkochen, Batengfurt, Germany) operated in secondary electron (SE) imaging mode. In order to evaluate crack propagation, the specimens perpendicular to the crack surface were vibro-polished without etching and were analyzed using an EVO 18 operated in electron backscattered diffraction (EBSD, Symmetry, Oxford Instruments plc, Tubney Woods, Abingdon, Oxfordshire, UK) imaging mode.

## 3. Results

### 3.1. Impact Toughness

Figure 3 shows the Charpy impact toughness of the CGHAZs of the HSLA steels with different Ca contents at −20 °C after large-heat-input welding at 400 kJ cm$^{-1}$. The impact toughness of each steel was tested three times, and the test values of the 11Ca and 18Ca steels revealed large variabilities. In order to evaluate these variabilities, two specimens each of the 11Ca and 18Ca steels were characterized, and they were denoted as 11Ca-1 and 11Ca-2, and 18Ca-1 and 18Ca-2 steels, respectively. The impact energies of the 11Ca-2 and 18Ca-2 steels are >100 J, much higher than the low values and similar to that of the

25Ca steel. However, the average impact energy of these HSLA steels increases with the increase in Ca content. It should be noted that the mechanical performance of CGHAZs is determined by the lowest value, guaranteeing the safe application of the 25Ca steel at $-20\,°C$.

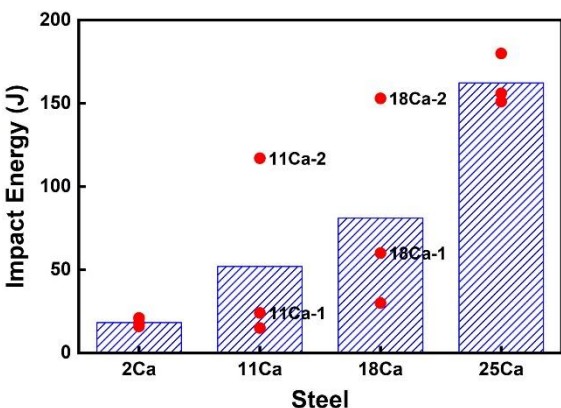

**Figure 3.** Charpy impact toughness of the CGHAZs of the HSLA steels at $-20\,°C$ with a welding heat input of $400\,kJ\,cm^{-1}$.

### 3.2. HAZ Microstructure

The CGHAZ grain structures of the HSLA steels were characterized, and they are shown in Figure 4a–f after large-heat-input welding at $400\,kJ\,cm^{-1}$. The grain boundary ferrite (GBF) is severely grown and coarsened on the prior austenite grain boundaries, along with the IAF formed within the grains. The grain sizes of the 2Ca and 11Ca-1 steels appear to be comparable, about 350 µm, a bit larger than that of the 18Ca-1 steel, which is measured to be 330 µm (Figure 4a,b,d). The grains of the 25Ca steel are significantly decreased to 290 µm, revealing a uniform and fine grain structure (Figure 4f). It is also noted that the average grain sizes of the 11Ca-2 and 18Ca-2 steels are similar to that of the 25Ca steel, but a small number of extremely large grains are also detected in the two steels (Figure 4c,e).

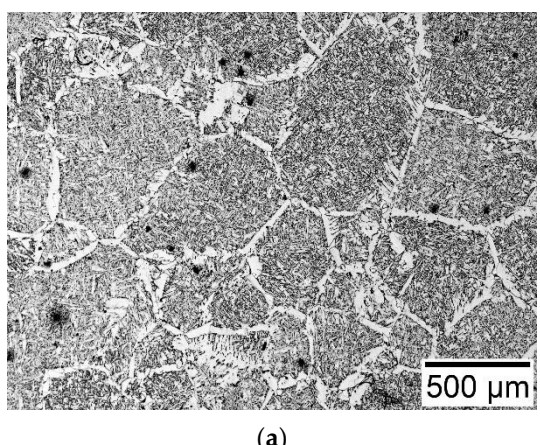
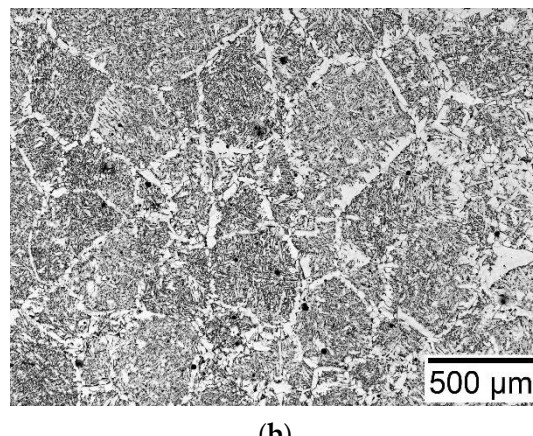

(**a**)                                                                                         (**b**)

**Figure 4.** *Cont.*

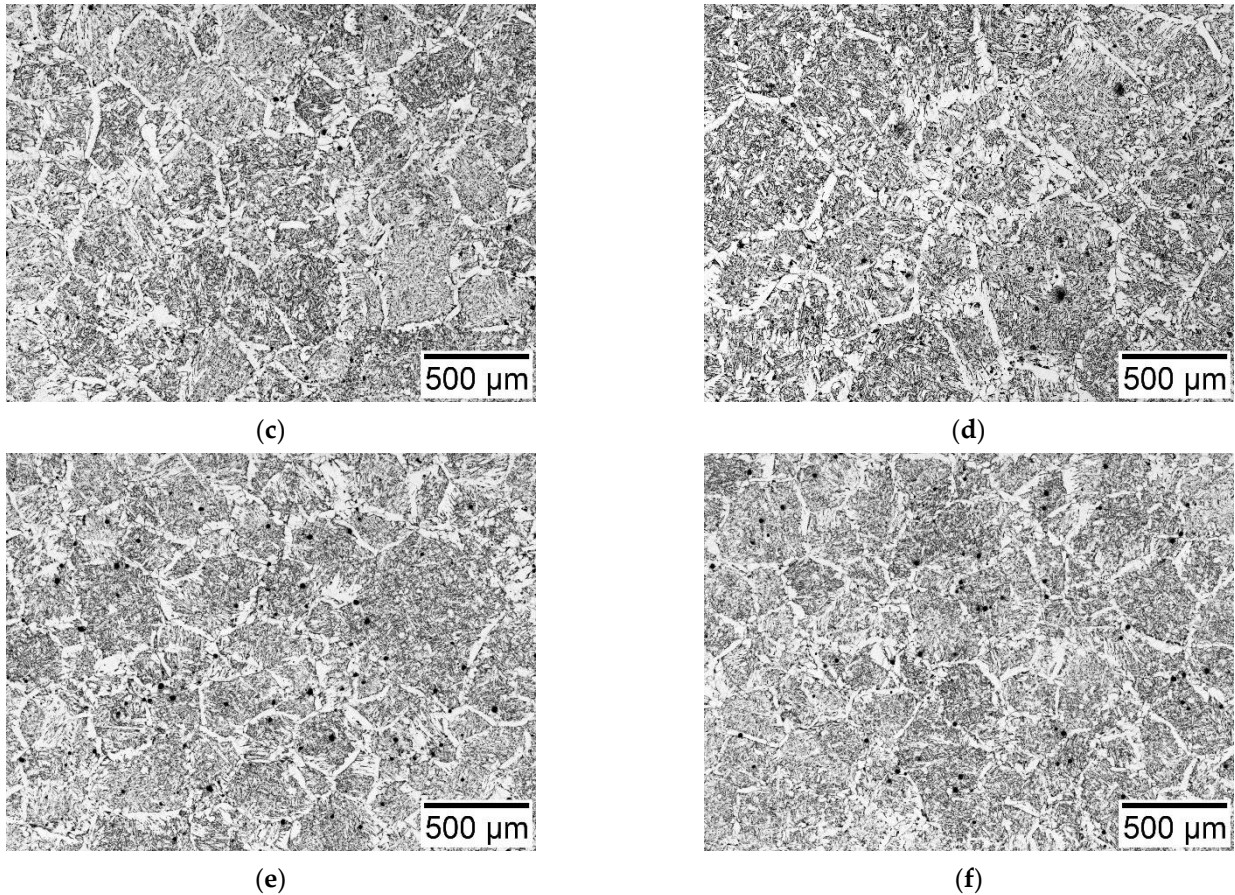

**Figure 4.** Optical images of the CGHAZ grain structures of (**a**) 2Ca, (**b**) 11Ca-1, (**c**) 11Ca-2, (**d**) 18Ca-1, (**e**) 18Ca-2 and (**f**) 25Ca steels after large-heat-input welding at 400 kJ cm$^{-1}$.

Figure 5 displays optical images showing the intragranular microstructures of the CGHAZs after large-heat-input welding at 400 kJ cm$^{-1}$. Fine and interlocking IAF is formed within the prior austenite grains of the CGHAZs of the HSLA steels regardless of the Ca content. The grain boundaries are completely enveloped by the coarse GBF, whose width is about 50 μm. It is interesting that a small amount of ferrite side plate (FSP) is observed in the CGHAZs of the 2Ca and 11Ca-1 steels, in which the prior austenite grains are relatively coarse. This FSP grows inwardly from the grain boundaries, showing a comb-like morphology.

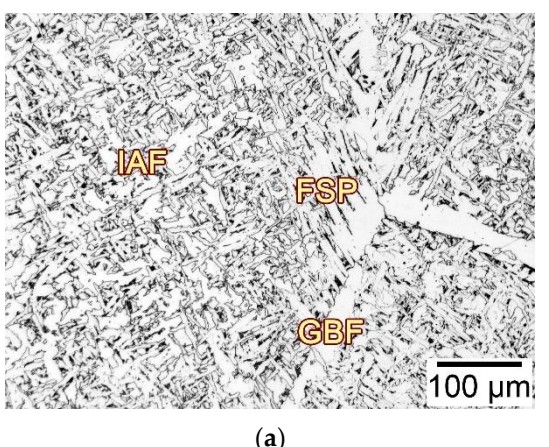

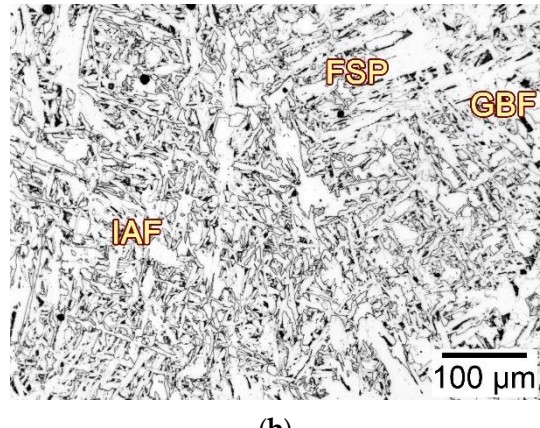

**Figure 5.** *Cont.*

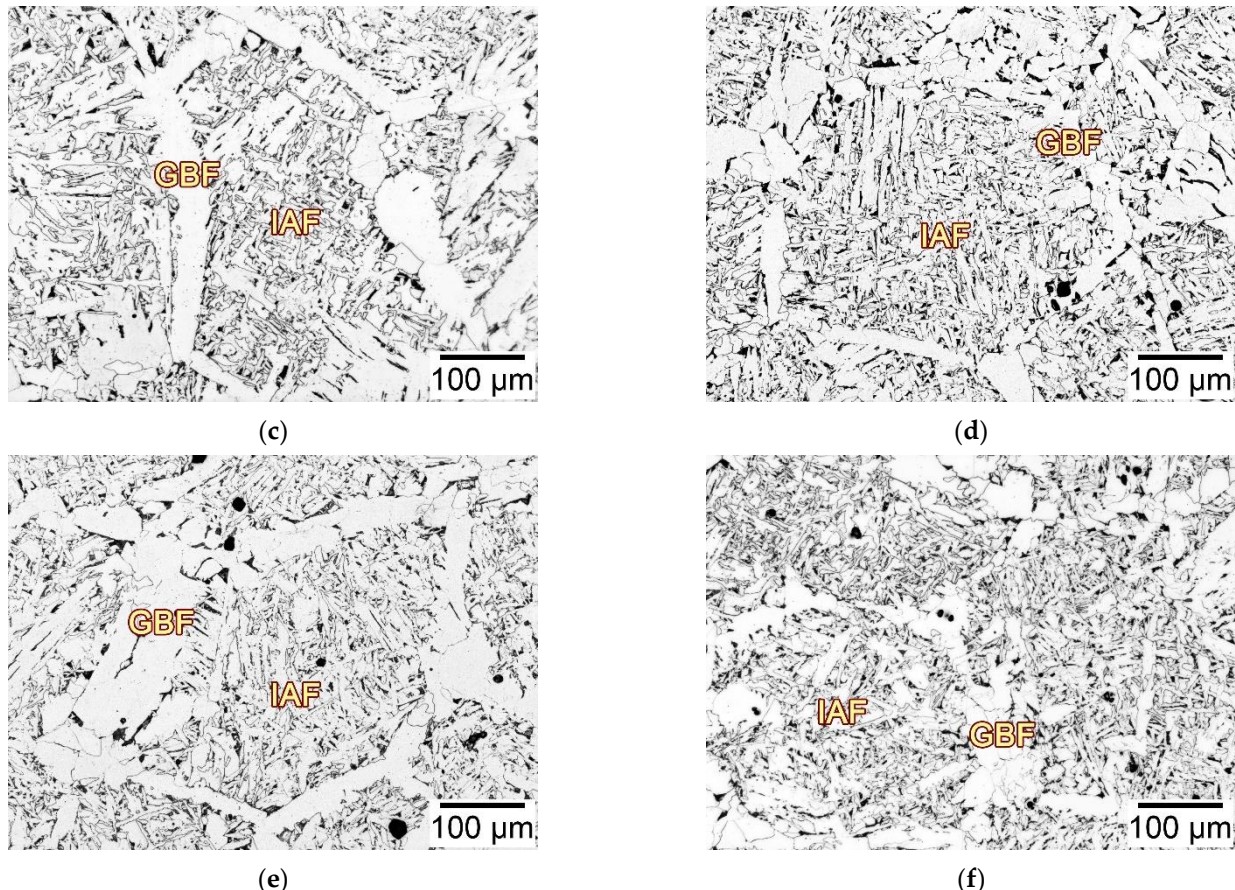

(c)

(d)

(e)

(f)

**Figure 5.** Optical images of the CGHAZ microstructures of (**a**) 2Ca, (**b**) 11Ca-1, (**c**) 11Ca-2, (**d**) 18Ca-1, (**e**) 18Ca-2 and (**f**) 25Ca steels after large-heat-input welding at 400 kJ cm$^{-1}$.

### 3.3. Fracture Surface

Figure 6 displays SEM-SE images showing the impact fracture surfaces of the CGHAZs of the HSLA steels with different Ca contents after large-heat-input welding at 400 kJ cm$^{-1}$. Figure 6a–e show the complete fracture surface of the CGHAZs, which consist of a fibrous zone, a radical zone and a shear lip zone. The V-notch and the fibrous zone are divided by the red line, the fibrous zone and the radical zone are divided by the green line, and the shear lip zone and the radical zone are divided by the blue line. It should also be noted that the fibrous zone does not appear during the impact fracture of the 2Ca and 11Ca-1 steels, as shown in Figure 6a,b.

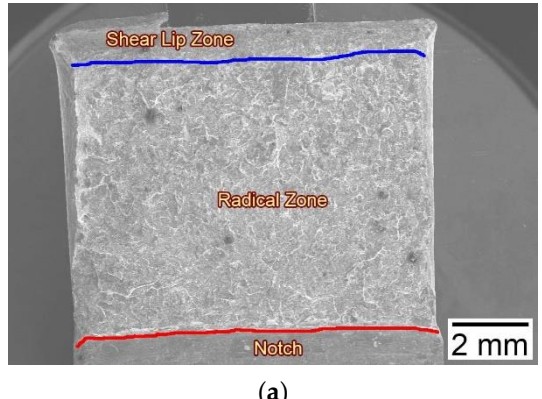

(a)

(b)

**Figure 6.** *Cont.*

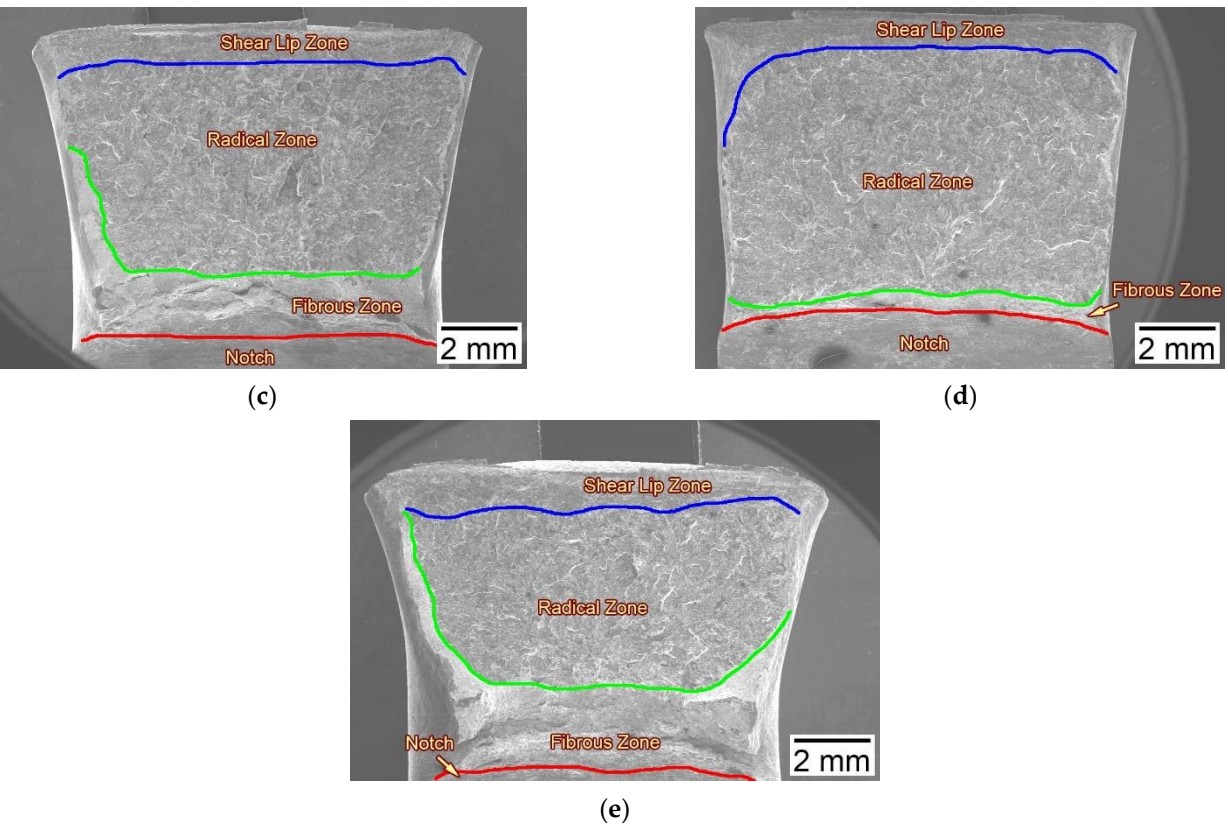

**Figure 6.** SEM-SE images showing the complete fracture surface of the CGHAZs of (**a**) 2Ca, (**b**) 11Ca-1, (**c**) 11Ca-2, (**d**) 18Ca-1 and (**e**) 18Ca-2 steels after large-heat-input welding at 400 kJ cm$^{-1}$.

The fracture surface of the 2Ca steel is dominated by the radical zone, the area of which is measured to be 67.2 mm$^2$, much larger than that of the shear lip zone, which is about 8.6 mm$^2$ (Figure 6a). The areas of the radical zone and the shear lip zone of the 11Ca-1 steel are measured to be 73.8 and 10.1 mm$^2$, respectively, comparable to those of the 2Ca steel (Figure 6b). In comparison with the 11Ca-1 steel, the area of the radical zone of the 11Ca-2 steel is decreased to 51.5 mm$^2$, while the area of the fibrous zone is increased to 15.4 mm$^2$, except for that of the shear lip zone, which is about 9.7 mm$^2$ (Figure 6c). The areas of the radical zone and the shear lip zone of the 18Ca-1 steel are measured to be 63.8 and 8.8 mm$^2$, comparable to those of the 2Ca and 11Ca-1 steels, but a small fibrous zone is also formed, the area of which is about 3.5 mm$^2$ (Figure 6d). The area of the radical zone of the 18Ca-2 steel is significantly decreased to 41.2 mm$^2$, while the area of the fibrous zone is clearly increased to 20.9 mm$^2$, except for the similar shear lip zone, which has an area of 10.8 mm$^2$ (Figure 6e).

The fibrous zone represents the ductile fracture, which is composed of dimples, and the radical zone indicates the brittle fracture, which is dominated by cleavage river patterns. The typical enlarged morphologies of the radical zones of the 11Ca-1 and 11Ca-2 steels are shown in Figure 7a,b, which indicate that dimples are present together with cleavage facets in the 11Ca-2 steel, different from that observed in the 11Ca-1 steel. The shear lip zone is the hammer-impacted zone, which is also composed of dimples, but the dimples are typically elongated, as shown in Figure 7c,d.

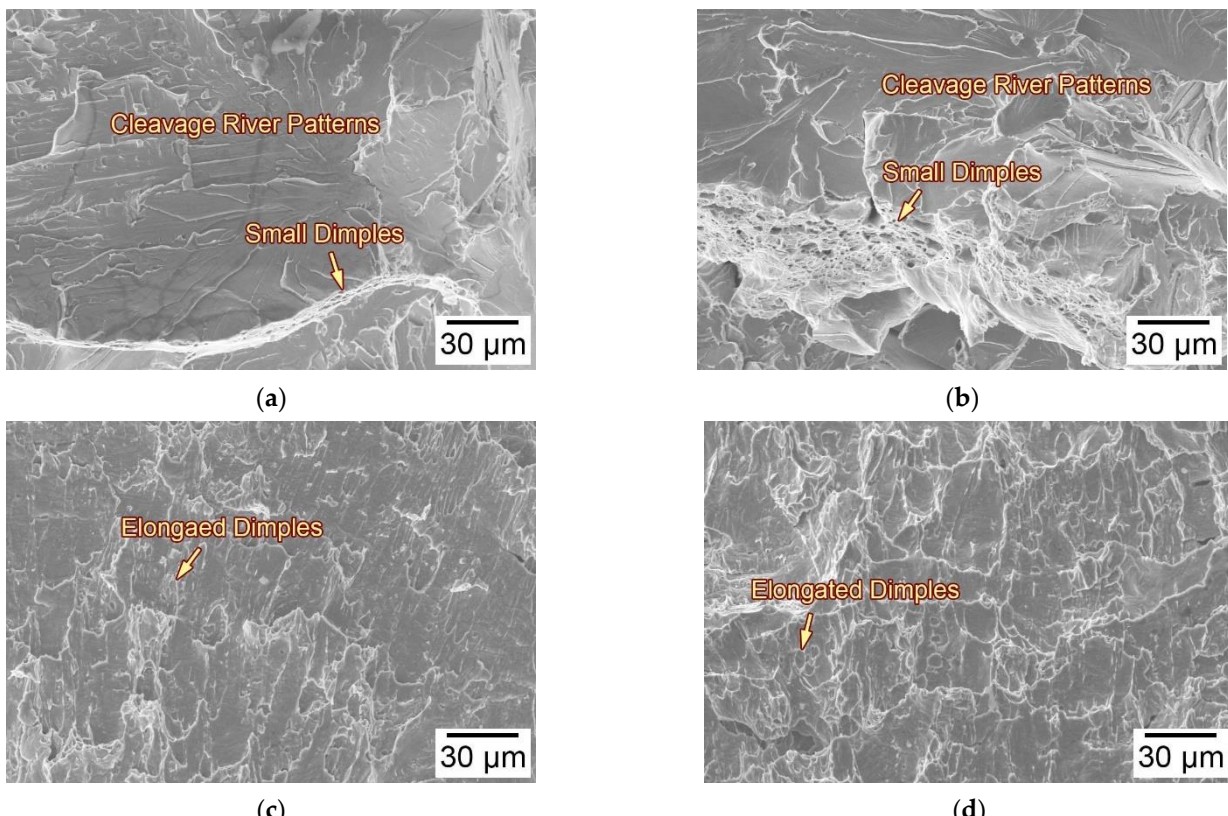

**Figure 7.** SEM-SE images showing the fracture surfaces of the (**a**,**b**) radical zone and (**c**,**d**) the shear lip zone in (**a**,**c**) 11Ca-1 and (**b**,**d**) 11Ca-2 steels after large-heat-input welding at 400 kJ cm$^{-1}$.

### 3.4. Secondary Cracks

In order to investigate the propagation of the impact cracks, the propagation of the secondary cracks near the fracture surface was investigated. Figure 8 shows the EBSD maps of the secondary cracks in the CGHAZs of the HSLA steels after large-heat-input welding at 400 kJ cm$^{-1}$. Figure 8a shows that the major crack in the 2Ca steel propagates along the coarse GBF, leaving straight fracture planes. The secondary crack also propagates along the GBF, with the crack direction perpendicular to the major crack. The propagation of this secondary crack is ultimately inhibited by the IAF.

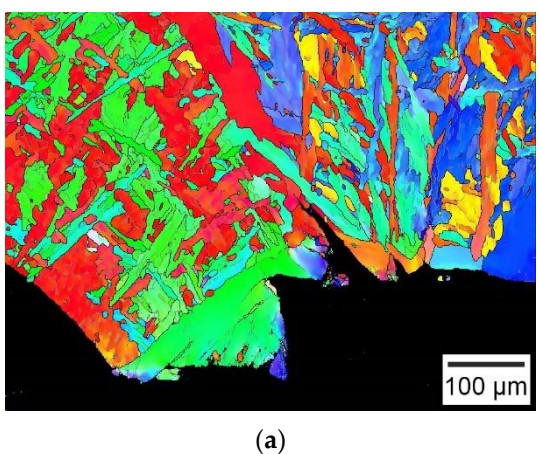

(**a**)

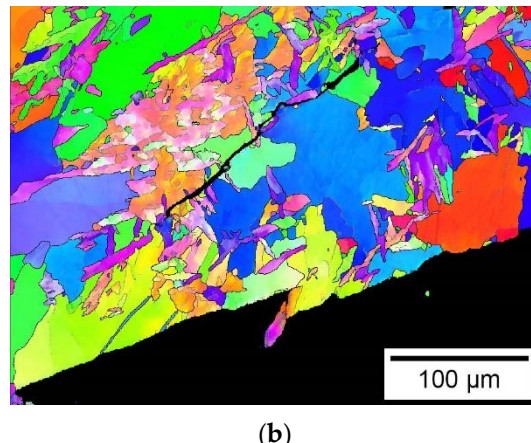

(**b**)

**Figure 8.** *Cont.*

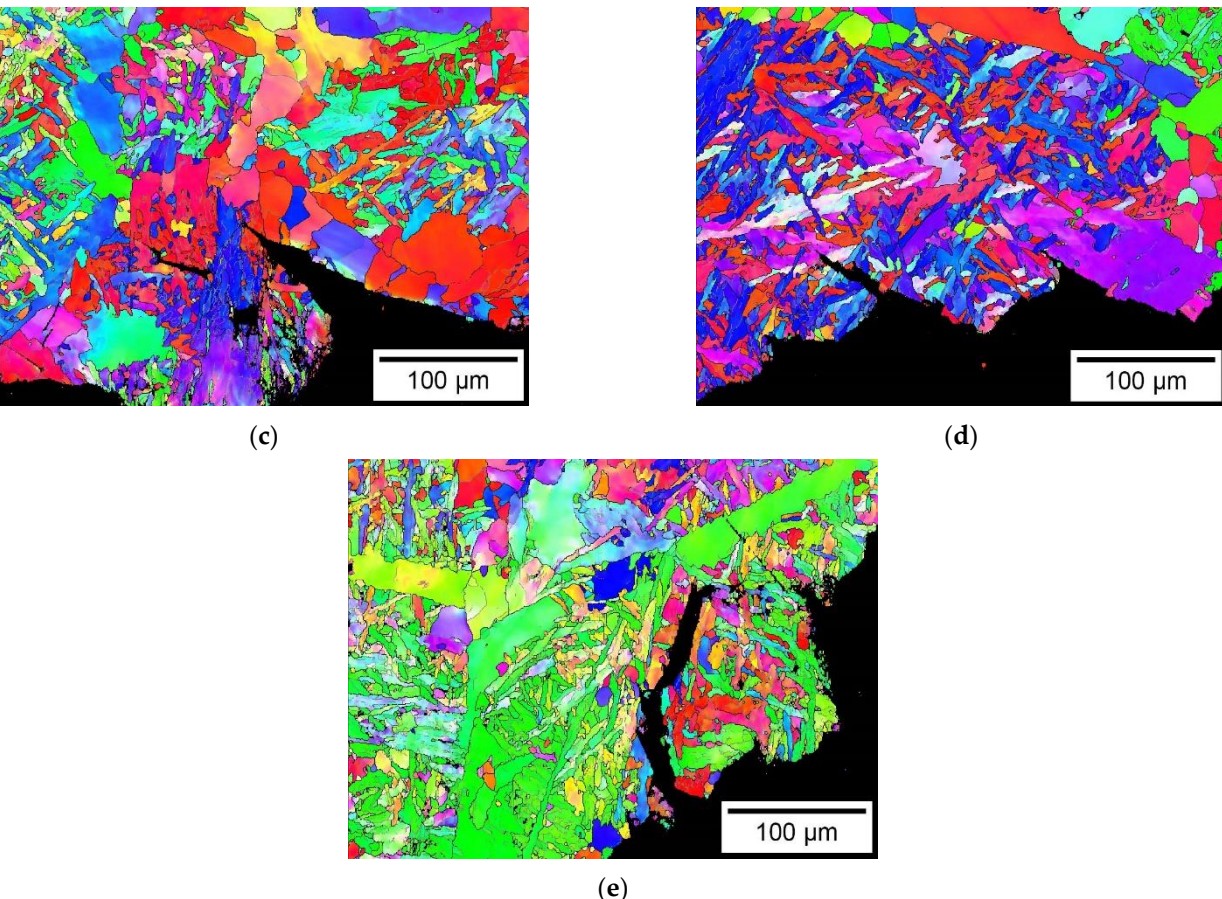

**Figure 8.** EBSD maps showing the crack propagation of the CGHAZs of (**a**) 2Ca, (**b**) 11Ca-1, (**c**) 11Ca-2, (**d**) 18Ca-1 and (**e**) 18Ca-2 steels after large-heat-input welding at 400 kJ cm$^{-1}$.

The major crack in the 11Ca-1 steel propagates along the coarse GBF, similar to that observed in the 2Ca steel (Figure 8b). The secondary crack nucleates on the coarse GBF, and it preferentially grows along the low-angle grain boundaries of the GBF, showing intergranular and transgranular cracks. The major crack in the 11Ca-2 steel also propagates along the GBF, but its propagation is inhibited by the IAF (Figure 8c). A secondary crack nucleates on the IAF, and its propagation is rapidly stopped by the high-angle boundaries of the IAF within the grain.

Figure 8d shows that the major crack in the 18Ca-1 steel propagates along the GBF and IAF, showing a serrated morphology. It is interesting that one secondary crack propagates along the GBF and that the other one grows along the IAF, but both of their propagations are stopped by the IAF. The major crack in the 18Ca-2 steel propagates along the IAF, also exhibiting a serrated morphology (Figure 8e). The secondary crack nucleates on the major crack and grows along the IAF, and its propagation is stopped by the high-angle boundaries before it reaches the GBF.

## 4. Discussion

The impact toughness of the CGHAZs of HSLA steels with various Ca contents is profoundly associated with prior austenite grains and intragranular structures [28]. The average impact toughness increases with an increase in Ca content, primarily attributed to Ca deoxidation, which improves the fine dispersion of TiN particles to prevent grain growth during the welding thermal cycle [19,21]. However, large variabilities in the impact toughness are observed in the HAZs of the 11Ca and 18Ca steels (Figure 3). The microstructural characterization demonstrates that the grain structures of the CGHAZs of the two steels are not uniform, and the size of the fine grains is about 40 μm smaller

than the coarse ones (Figure 4b–e). As a result, the HAZ samples of the 11Ca-2 and 18Ca-2 steels with fine grains exhibit high performances, similar to that of the 25Ca steel, while the samples with coarse grains in the 11Ca-1 and 18Ca-1 steels are extremely brittle.

This nonuniform distribution of the prior austenite grains is primarily ascribed to the nonuniform distribution of the sub-micron scale TiN particles during steelmaking. Our previous study summarized that the number density of TiN particles increases with an increase in Ca content, and the particle size is significantly decreased in the 25Ca steel with a relatively high Ca content [21]. This suggests that, although the quantity of TiN particles increases in the 11Ca and 18Ca steels, the quantity is still not high enough to obtain a uniform dispersion in these steels, unlike that in the 25Ca steel. As a result, the nonuniform dispersion of the TiN particles refines a part of the grain structure of the CGHAZs of the 11Ca and 18Ca steels, leading to large impact test variabilities.

In order to elucidate the fracture mechanism, the Ca effect on the TiN particles, the microstructure and the impact toughness of the CGHAZs are schematically exhibited in Figure 9. Our previous studies claimed that the combined deoxidation of Ti and Ca can effectively induce the formation of strong IAF and inhibit the formation of brittle FSP, and this is beneficial to prevent crack propagation [20,21]. Additionally, the size and uniformity of prior austenite grains and the quantity of GBF affect HAZ toughness more profoundly than the intragranular structures of these HSLA steels.

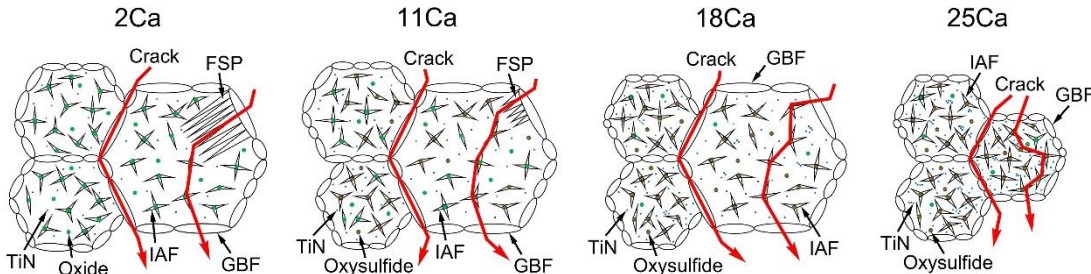

**Figure 9.** Schematic diagram of microstructural evolution and impact crack path of CGHAZ of the HSLA steels with the variation in the Ca content. IAF means intragranular acicular ferrite. GBF means grain boundary ferrite. FSP means ferrite side plate.

Based on the combined analyses of the fracture surfaces and the CGHAZ structures of the 2Ca, 11Ca-1 and 18Ca-1 steels (Figures 4 and 6), it is concluded that the relatively coarse grains accelerate crack propagations by forming large radical zones, thus degrading the impact toughness at −20 °C. Different from these three steels, the brittle radical zones of the 11Ca-2 and 18Ca-2 steels are significantly narrowed by the expansion of the ductile fibrous zones, indicating the retarding of the crack propagations by these ductile areas. This is primarily ascribed to the small prior austenite grains of the CGHAZs that enforce the swerve of the crack propagation, similar to the crack path of the 25Ca steel in Figure 9. The appearance of the small fibrous zone in the 18Ca-1 steel also suggests that its impact toughness increased by a small amount, owing to the grains being finer than those in the 2Ca and 11Ca-1 steels (Figure 6d).

The grain boundaries of the prior austenite grains are completely enveloped by the coarse GBF with low-angle boundaries (<15°), with extremely low resistance to crack propagation, but in contrast, the IAF with high-angle boundaries within the grains contributes to the tortuous paths and can even arrest cracks at boundaries with misorientations larger than 50° [29]. Microstructural characterization from the longitudinal direction of the fracture surface confirms that the major and secondary cracks in the steels with coarse grains (the 2Ca and 11Ca-1 steels) preferentially propagate along the coarse GBF, resulting in extremely low HAZ toughness (Figure 8a,b). The major and secondary cracks begin to grow along the IAF in the 11Ca-2 and 18Ca-1 steels, indicating that the relatively fine grains and the decreased amount of GBF effectively prevent crack propagation (Figure 8c,d).

The proportion of the major and secondary crack paths along the IAF in the 18Ca-2 steel increases significantly, suggesting that the uniform and fine grains can effectively retard the crack propagation and increase HAZ toughness (Figure 8e). The HAZ toughness of the 25Ca steel is excellent, predominantly owing to the uniform and fine grains that effectively prevent the cracks from growing.

It is worth noting that safe application can only be guaranteed for the 25Ca steel at $-20\ ^\circ$C and not the 11Ca and 18Ca steels, owing to the high HAZ toughness and the small test variability. Because of the positive effect of Ca deoxidation on the refinement of prior austenite grains, it is strongly recommended to increase the Ca content to about 25 ppm in this type of HSLA steel system. A relatively high Ca addition can also increase the uniformity of the grains, thereby decreasing test variability.

## 5. Conclusions

The HAZs of a series of HSLA steels with different Ca contents were simulated for large-heat-input welding at 400 kJ cm$^{-1}$, and the effect of Ca content on the impact crack behavior of the simulated CGHAZs was investigated. The following conclusions were drawn:

1. The average impact toughness of CGHAZs increases with an increase in Ca content, and the HAZ toughness of the 25Ca steel is satisfactorily high, primarily owing to the uniform and fine prior austenite grains.
2. The prior austenite grains in the CGHAZs of the 11Ca and 18Ca steels with relatively low Ca contents are not uniform, leading to large test variabilities in HAZ toughness at $-20\ ^\circ$C.
3. The major and secondary cracks in the impact tests preferentially propagate along the brittle GBF, but the cracks alter to grow along the strong IAF as the grain size decreases, resulting in improved HAZ toughness.

**Author Contributions:** Y.Z. (Yinhui Zhang) and J.Y. conceived and designed the study. Y.Z. (Yinhui Zhang) and H.D. performed the experiments. Y.Z. (Yinhui Zhang) analyzed the experimental results and wrote the manuscript. J.Y., Y.Z. (Yu Zhang) and H.M. contributed to English correction and proofreading. All authors have read and agreed to the published version of the manuscript.

**Funding:** This work was financially supported by the National Natural Science Foundation of China (No. U1960202); the Independent Research and Development Project of State Key Laboratory of Advanced Special Steel, Shanghai Key Laboratory of Advanced Ferrometallurgy, Shanghai University (No. SKLASS 2021-Z09); and the Science and Technology Commission of Shanghai Municipality (No. 19DZ2270200 and 20511107700).

**Data Availability Statement:** Data are contained within the article.

**Acknowledgments:** The authors are grateful to X.F. Jiang and R.Z. Wang for their assistance in steelmaking and TMCP treatment.

**Conflicts of Interest:** The authors declare no conflict of interest.

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
