# Peer review of "Effect of Ca Deoxidation on Toughening of Heat-Affected Zone in High-Strength Low-Alloy Steels after Large-Heat-Input Welding"

_metals, doi:10.3390/met12111830_

Round 1

Reviewer 1 Report

The paper analyses large heat input welding can effectively increase the efficiency and reduce the cost of manufacturing a super-large container ship for marine trade worldwide with thick high-strength low-alloy (HSLA) steel plates, whereas it significantly degrades the toughness of the welding heat affected zone (HAZ). This paper describes the effect of Ca deoxidation on the impact toughness of the simulated coarse-grained HAZ (CGHAZ) in the HSLA steels after large heat input welding at 400 kJ cm-1. The average impact energy of the CGHAZ increases with increasing Ca content, and particularly the energy of the steel with 25 ppm Ca content is satisfactorily high, owing to the uniform and fine prior austenite grains. In contrast, the grains in the CGHAZ of the steels with relatively low Ca contents are not uniform, leading to large test variabilities at -20 oC. Failure analyses reveal that the major and secondary cracks preferentially propagate along the coarse and brittle grain boundary ferrite (GBF), but their propagation will be retarded by the fine and inter-locking intragranular acicular ferrite (IAF) as the grain size is decreased. It is therefore recommended to increase the Ca content during deoxidation of steelmaking for the related HSLA steels.

I recommend paper for printing and publishing 

Author Response

Point 1: The paper analyses large heat input welding can effectively increase the efficiency and reduce the cost of manufacturing a super-large container ship for marine trade worldwide with thick high-strength low-alloy (HSLA) steel plates, whereas it significantly degrades the toughness of the welding heat affected zone (HAZ). This paper describes the effect of Ca deoxidation on the impact toughness of the simulated coarse-grained HAZ (CGHAZ) in the HSLA steels after large heat input welding at 400 kJ cm-1. The average impact energy of the CGHAZ increases with increasing Ca content, and particularly the energy of the steel with 25 ppm Ca content is satisfactorily high, owing to the uniform and fine prior austenite grains. In contrast, the grains in the CGHAZ of the steels with relatively low Ca contents are not uniform, leading to large test variabilities at -20 oC. Failure analyses reveal that the major and secondary cracks preferentially propagate along the coarse and brittle grain boundary ferrite (GBF), but their propagation will be retarded by the fine and inter-locking intragranular acicular ferrite (IAF) as the grain size is decreased. It is therefore recommended to increase the Ca content during deoxidation of steelmaking for the related HSLA steels.

I recommend paper for printing and publishing

Response 1: Thank you so much for your comments.

Reviewer 2 Report

The paper is well written and comprehensive in its theme. Minor revisions is requested, concerning the specification of parameters of the Electrogas welding procedure simulated and identification of all the acronyms used in the text.

Reviewer 3 Report

Review report on the topic ‘Effect of Ca deoxidation on toughening of heat-affected zone in high-strength low-alloy steels after large heat input welding’. The work is presented well. The comments to improve the quality of the manuscript are listed below:

  1. Please omit the unnecessary information and add the key conclusion of the work at the end of the abstract section.
  2. Please add a separate section to discuss the novelty of the work.
  3. The introduction section is presented roughly. Add more references and try to make a bridge between current and previously published work and also refer the paper published on CGHAZ simulation.
  4. : https://doi.org/10.1007/s11665-021-06177-2; https://doi.org/10.1016/j.cirpj.2021.09.002.
  5. How was the weight percentage of the Ca was selected?
  6. Is there any test performed experimentalluy? If yes, add the image of eth tested specimen.
  7. What was the standard used for imopact tested specimen in simulation work. Add the clear location in schematic image along with diemnsions.
  8. The HAZ microstructure was presented of very poor quality and needs major attention. Add good quality image along with technical discussion also add the grain size measurement and discuss the effect of the microstructure on impact toughness value.
  9. The fracture surface study needs a major correction. Add technical discussion. Mark the region of brittles and the ductile area along with EDS spectra of eth fracture surface: https://doi.org/10.1016/j.engfailanal.2018.09.036.
  10. A discussion section is presented well but needs more references in support of statements.

Reviewer 4 Report

The work concerns the influence of Ca on changes in the HAZ structure of HSLA steels during simulated electrogas welding. The paper does not precisely describe why electrogas welding was simulated for welding HSLA steels and why the linear welding energy was adopted at the level of 400 kJ / cm. Is this type of technology used for welding container ships? On what basis was the cooling time in the HAZ assumed to be 385 s? The conducted research is basic and its full interpretation requires additional structural research. Why was the EBSD method used in the study to interpret the structure if the grain orientation and angular boundaries were not analyzed? In my opinion, the article should not be published at this stage, after verifying the scope and completing the results and their analysis, it can be reviewed.

Author Response

Response to Reviewer 4 Comments

Point 1: The work concerns the influence of Ca on changes in the HAZ structure of HSLA steels during simulated electrogas welding. The paper does not precisely describe why electrogas welding was simulated for welding HSLA steels and why the linear welding energy was adopted at the level of 400 kJ / cm.

Response 1: Thank you for your comments. The electrogas welding method is a high performance welding method that applies two welding torches in the plate thickness direction with the one pass automatic welding method developed by the Nippon Steel Welding Products & Engineering Co., Ltd., and uses a flux wire in a swinging copper plate electrode, and a solid wire in the back side electrode, and uses a dedicated back material. This on pass welding demands the heat input as high as 400 kJ/cm.

Line 85~87: The electrogas arc welding with the heat input of 400 kJ cm-1 is a one pass automatic welding method, which can significantly increase the welding efficiency and reduce the cost [27].

Point 2: Is this type of technology used for welding container ships?

Response 2: Thank you for your comments. This welding technology is used for container ships.

Point 3: On what basis was the cooling time in the HAZ assumed to be 385 s?

Response 3: Thank you for your comments. The heating temperature and time is recorded by Gleeble 3800, and thus the cooling time can be calculated. The thermal cycle is added in the revised version, as shown in Figure 2.

Point 4: The conducted research is basic and its full interpretation requires additional structural research. Why was the EBSD method used in the study to interpret the structure if the grain orientation and angular boundaries were not analyzed? In my opinion, the article should not be published at this stage, after verifying the scope and completing the results and their analysis, it can be reviewed.

Response 4: Thank you for your comments. It is convenient to interpret the crack propagation in the CGHAZ by use of the EBSD method. In our previous, we have found that the coarse GBF with low-angle boundaries (< 15o) is extremely low resistant to crack propagation, but in contrast, the IAF with high-angle boundaries within grains con-tributes to the tortuous paths and can even arrest cracks at boundaries with misorien-tations larger than 50o, and based on this result, the failure behaviour is discussed. Our previous paper has been referred in the current research.

Pan, X.Q.; Yang, J.; Zhang, Y.H. Microstructure and fracture characteristics of heat-affected zone in shipbuilding steel plates with Mg deoxidation after high heat input welding. Steel Res. Int. 2021, 92, 2100376.

Round 2

Reviewer 4 Report

After the corrections and additions are made, the article can be published